# Temporal Pattern Analysis of Local Rainstorm Events in China During the Flood Season Based on Time Series Clustering

**Fan Wang** [1,2] 

[1] State Key Laboratory of Hydroscience and Engineering, Department of Hydraulic Engineering, Tsinghua University, Beijing 100084, China; wangfan@iwhr.com

[2] State Key Laboratory of Simulation and Regulation of Water Cycle in River Basin, Research Center on Flood & Drought Disaster Reduction of the Ministry of Water Resources, China Institute of Water Resources and Hydropower Research, Beijing 100038, China

**Abstract:** Similar to the rainfall depth, duration and intensity, the temporal pattern is also an important characteristic of rainstorm events. Studies have shown that temporal patterns will influence runoff modelling, flash flood warning thresholds as well as urban and infrastructure flood inundation simulations. In this study, a time series clustering method using dynamic time warping (DTW) as similarity measurement criteria is proposed to analyze rainfall temporal patterns. Compared with the existing approaches, it can better reflect the real rainfall processes. Based on this novel method, five representative temporal patterns were extracted from 13,299 rainstorm events during the flood season in China. Through the analysis of their statistical characteristics, the disaster-causing risks of each temporal pattern were compared. Furthermore, we found that for rainstorm events whose durations are less than 24 h, the rainfall is mainly concentrated in 3 to 6 h, which proposes higher requirements for the design of flood control and drainage projects compared with those using average intensities of 12 or 24 h as design standards. Finally, through regional analysis, we found that the rainfall depth, intensity and peak value are affected by the macroclimate. However, the temporal patterns are not strongly related to the macroclimate but are more likely to be affected by the local climate and topography, which needs further studies at smaller scales.

**Keywords:** temporal pattern; local rainstorm events; flood season; time series clustering; Dynamic Time Warping (DTW)

---

## 1. Introduction

Intense local rainstorm events are deemed to be the primary triggering factor of flash flooding and urban flood inundation [1,2]. When describing the characteristics of rainstorm events, the rainfall depth, duration and intensity are generally adopted; a commonly overlooked element is the temporal pattern. The temporal pattern of rainstorm events refers to the distribution of rainfall intensity within the event, which reflects the process of the occurrence, development and extinction of rainfall events [3]. Although the main features of flood and inundation events are generally controlled by rainfall intensity and duration, the processes of events are still closely related to the temporal patterns of rainstorms, which has already been illustrated by current studies. The main concerns of the previous literature have focused on the influence of rainstorm temporal patterns on flood runoff modelling and analysis [4–7], flash flood warnings [8–12] and urban and infrastructure flood inundation simulation [2,13,14].

By applying a GIS-supported GIUH approach, Jain and Singh [5] observed that the peak characteristics of the design flood are sensitive to the temporal patterns of rainstorms. Máca and

Torfs [6] confirmed the impact of temporal rainfall distribution on flood runoff by performing designed evaluations. Lin and Chen [7] found that the time variation coefficient of rainstorms had a significant influence on flood peak value by regression analysis. Rahman and Weinmann [15] and Rahman and Islam [16] proposed a design flood estimation technique considering temporal patterns based on joint probability principles, which realized a more precise reproduction of the observed frequency curves.

Since the response time of flash floods is commonly under a few hours, the lead time is quite short for hydrological forecasting [1]; therefore, the most widely accepted approach for flash flood warnings is to establish rainfall thresholds [17–20], of which a representative one is the Flash Flood Guidance (FFG) developed by the US Hydrologic Research Center [21–23]. Previous studies have shown that the initial soil moisture conditions and rainfall temporal pattern have great impacts on determining the rainfall threshold [6,8,10,24]. Forestieri and Caracciolo [9] analyzed different rainfall thresholds by considering various synthetic hyetograph types. Zhai and Guo [12] found that the effect of rainfall temporal patterns on rainfall thresholds increased from advanced, intermediate to delayed patterns. Yuan and Liu [11] had similar results and argued that the rationality of the rainfall pattern should be considered in determining the rainfall threshold.

In previous urban inundation simulations and analyses, the rainstorm temporal pattern was commonly generalized as uniform or triangle, and only a single design pattern was taken into consideration [14]. Current studies revealed that as the direct cause of inundation, the distribution of rainstorm intensity within the event is highly related to the maximum range and depth of inundation areas. The study of Hou and Guo [2] illustrated the quantitative relationship between different rainstorm temporal patterns and unban inundation disasters by numerical simulation. Li and Wu [13] found that the reduction effects of various LID measures on the runoff volume and peak flow are different with the impacts of rainfall temporal patterns.

In current studies concerning the influence of rainfall temporal patterns, synthetic rainfall distributions of design storms have been widely used [25,26], which are generally recommended in regional rainstorm handbooks in China. A considerable number of approaches have been proposed to analyze synthetic rainfall distributions [27–32], mainly including the following methodologies: (a) derived from the intensity-duration-frequency (IDF) relationship and rainstorm intensity formula [27]; (b) by considering the dimensionless first statistical moment of a practical and assumed hyetograph [31]; (c) by analyzing normalized cumulative distribution curves based on probability statistics [28,29]; and (d) directly establishing temporal patterns based on the statistics of historical rainstorm events [30]. The methods mentioned above are the most classic and widely applied ones, and various novel and improved approaches have been proposed in recent years. Wu and Yang [33] identified the rainfall temporal patterns by analyzing dimensionless rainfall patterns with statistical cluster methods. Terranova and Iaquinta [34] improved Huff's curves by proposing a new criterion for classifying standardized rainfall profiles. Ghassabi and Kamali [35] computed the rainfall temporal patterns based on a three-parametric logistic function. Jun and Qin [36] established a representative quartile of the design storm by considering inter-event time definition in Huff's method.

Despite the wide applications, there are still many differences between synthetic rainfall patterns and actual rainfall processes: (a) the durations of some design rainfalls are fixed, such as 24 h for the SCS rainfall pattern, but the durations of practical rainstorms can range from hours to several days; (b) the use of dimensionless cumulative curves, as in Huff's method, makes it possible for the comparison between events, but it seems irrational to treat the cumulative processes of rainstorms with durations of a few hours and several days as the same; (c) the synthetic rainfall patterns are mostly generalized, smooth curves or envelope curves, such as the patterns proposed by Keifer and Chu [27] and Yen and Chow [31], but the actual intensity distributions of rainstorms can be quite diverse and even include short intervals within the processes; (d) some of the methods are based on statistical analysis but mainly focus on the position of the peak or by means of calculating the first statistical moment, which makes it hard to master the overall process of rainstorm events. The differences mentioned above make the synthetic rainfall patterns less representative of local rainstorm events.

Time series similarity analysis and clustering methods are potential tools for analyzing the temporal pattern of local rainstorm events. First, time series similarity analysis has the ability to identify the overall features of rainstorm events, including the occurrence, development and extinction. Furthermore, appropriate similarity measurement criteria, such as dynamic time warping (DTW), make it possible to compare events with different durations without normalization. After that, clustering analysis can achieve the extraction of the representative rainfall pattern from a group of similar events. The objective of this study is to categorize rainstorm events during the flood season in China into several time sections based on their durations and identify the representative temporal patterns of each duration section using time series clustering methods.

## 2. Materials and Methods

### 2.1. Rainfall Data

Hourly rainfall data from 99 hydrological and precipitation stations managed and operated by the Ministry of Water Resources of China (MWR) were collected for the analysis. The time range of the rainfall data was from 2011 to 2018. Only data during flood seasons, which are basically from April to October, with slight differences between regions, were used. The stations cover latitudes from 22.9° N to 47.9° N, cover longitudes from 94.3° E to 132° E, and are located in seven geographical zones of China (Figure 1).

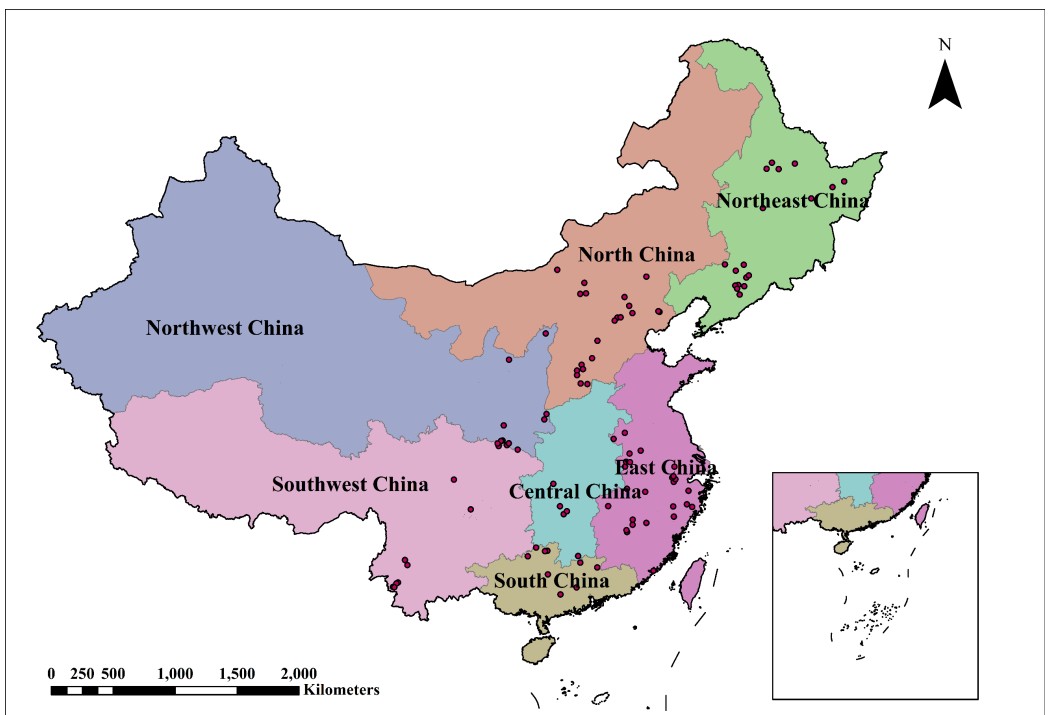

**Figure 1.** Distribution of the stations.

Individual rainstorm events were extracted from the continuous rainfall time series data. Same as the principle of Huff [27], two adjacent events were divided by a non-precipitation interval period of at least 6 h. As the distribution of rainfall intensity within the event is the main research object, events with durations of less than 3 h were excluded. In addition, rainstorms with total rainfall depths less than 10 mm were regarded as trace rainfalls and were not taken into consideration. As a result, 13,299 rainstorm events were recognized from the original rainfall data, with the number of events ranging from 782 to 4958 among the zones (Table 1). The distribution of rainstorm event durations ranged from 3 to 189 h, as shown in Figure 2.

**Table 1.** Overview of the rainfall data.

| Geographical Zones | Latitude and Longitude Range of Stations | Scope of Flood Season | No. of Stations | No. of Events |
|---|---|---|---|---|
| Northeast China | 41.1° N–47.9° N 123.4° E–132° E | June–September | 18 | 1255 |
| North China | 36.1° N–42.5° N 111.2° E–118.6° E | June–September | 21 | 1384 |
| Central China | 25° N–30.1° N 110.9° E–112.9° E | April–October | 6 | 1150 |
| East China | 24.5° N–33.2° N 114.9° E–121° E | April–September | 26 | 4958 |
| South China | 22.9° N–26° N 109.1° E–114.1° E | April–October | 8 | 2027 |
| Northwest China | 32.4° N–39° N 107.1° E–110.4° E | June–October | 9 | 782 |
| Southwest China | 23.4° N–32.5° N 99.3° E–108.3° E | May–October | 11 | 1743 |

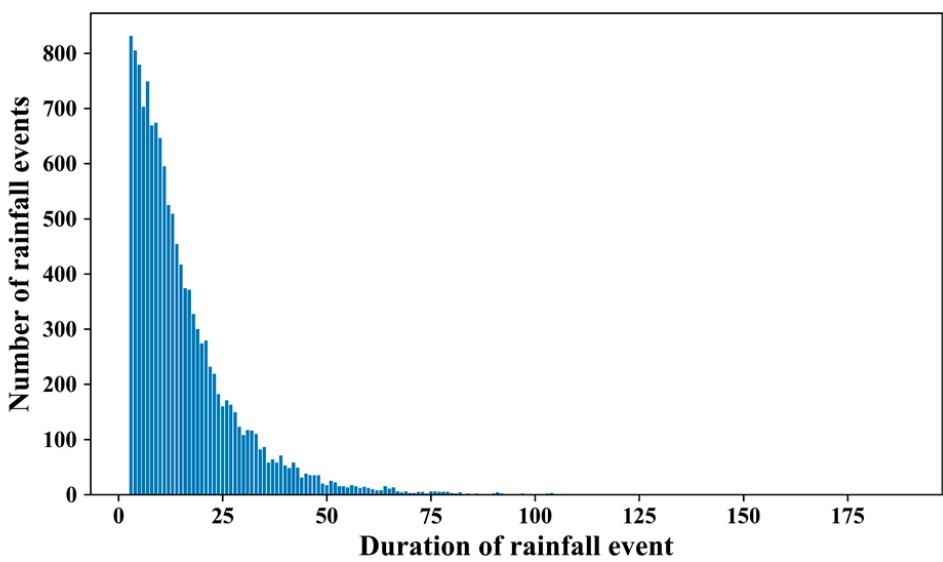

**Figure 2.** Distribution of rainstorm event durations.

*2.2. Methods*

The proposed temporal pattern analysis method based on time series clustering includes the following three procedures: the standardization and grouping of rainfall time series, time series clustering and the extraction of representative temporal patterns based on clustering trees.

2.2.1. Standardization and Grouping of Rainfall Time Series

To eliminate the effects of rainfall magnitude in temporal pattern analysis, standardized rainfall cumulative curves were used. The following formula was used to standardize the rainfall process:

$$P_i' = \frac{\sum_{j=1}^{i} P_j}{\sum_{j=1}^{n} P_j} \tag{1}$$

where $n$ is the length of the rainfall process and $P_i$ is the rainfall intensity in the $i$th time step. As the durations of actual rainstorms may range from a few hours to hundreds of hours, it is not rational

to compare all the events in one group by generalizing the time dimension. The actual time series durations of rainstorms were retained in this study to ensure that the representative temporal patterns were more approximate to the actual process, and the total events were divided into groups based on event durations. Since the range of rainfall duration is from 3 to 189 h, the duration sections were set as [3, 6), [6, 12), [12, 24), [24, 48), [48, 96) and [96, 192) respectively; therefore, the total rainstorm events were divided into six groups, and the representative temporal patterns of rainstorms were extracted based on each group. The distribution of events and the number of events in each group are shown in Figure 3.

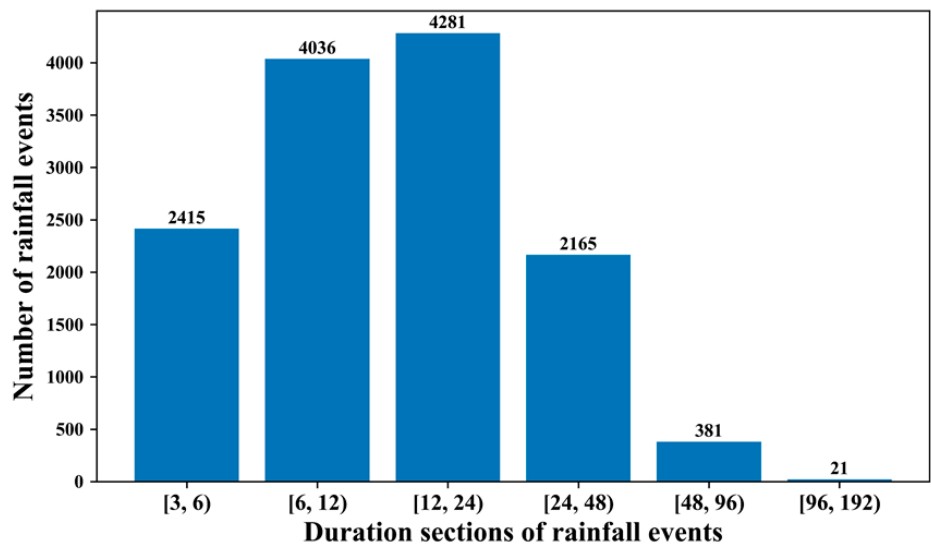

**Figure 3.** Distribution of rainstorm events in duration sections.

2.2.2. Time Series Clustering

Time series clustering is the process of dividing a time series set into multiple subsets (clusters) according to certain standards or rules (similarity measurement criteria), which aims to achieve a high similarity of elements within clusters and a low similarity of elements between clusters.

The first key factor of time series cluster analysis is the similarity measure criteria, which implies how to evaluate the similarity degree of two different time series. The choice of the similarity measurement criteria can determine the effect of clustering. At present, the similarity measurement of time series mainly depends on distance-based criteria, such as the Minkowski distance, Euclidean distance, Manhattan distance and dynamic time warping (DTW), etc. Among these criteria, the Euclidean distance and DTW are the most widely applied ones [3,33,37–41]. DTW first appeared as a spoken word recognition algorithm based on dynamic programming [42]; it solved the obstacle of pattern matching with different pronunciations, and was then introduced into the similarity measurement of time series [37,38]. Compared with the Euclidean distance, DTW supports the comparison between unequal time series, and can handle the measurement of complex time series with stretching or shifting on the time dimension. Two assumed rainstorm events were taken to illustrate the different similarity measurements between Euclidean distance and DTW. The two rainfall time series were set as [0.5, 3, 1, 2, 4, 6, 2.5, 0.2, 0.5, 0.2, 0.2] and [0.2, 0.5, 0.2, 2, 0.9, 1.5, 3, 5, 2, 0.2, 0.2], both with durations of 11 h, as shown in Figure 4.

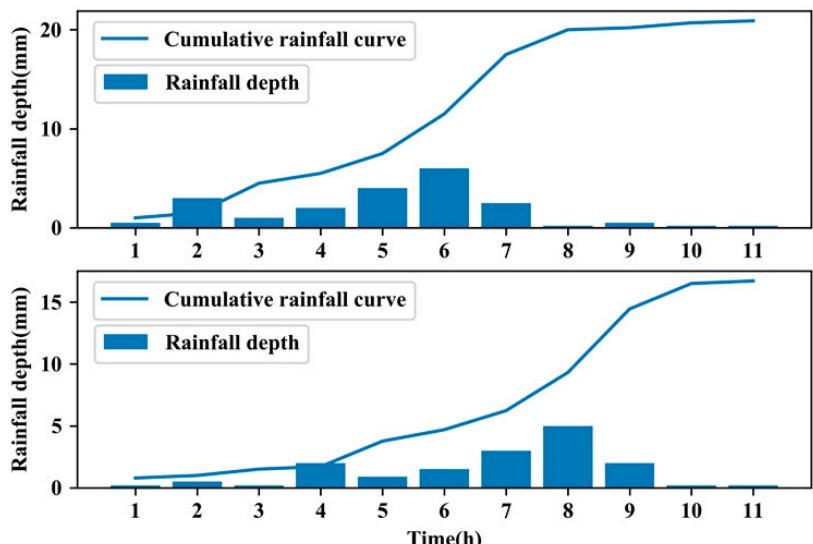

**Figure 4.** Sketch map of rainfall processes.

An intuitive comparison of the similarity measurement of the cumulative rainfall curves by Euclidean distance and DTW is shown in Figure 5. The Euclidean distance is based on the comparison of corresponding points of each time step, yet the DTW seeks a warping path that allows the comparison of one-to-many points. It is obvious that even though there exists a shifting of 2 h in the time dimension, the whole processes of the two rainfall processes are basically the same. The calculated Euclidean distance of the two cumulative time series is 20.0, which is much greater than the calculated DTW distance, which is 1.1. The comparison shows that DTW is more suitable for the evaluation of the similarity of rainfall processes because it can filter out the influence of trace rainfall before and after the main rainfall process.

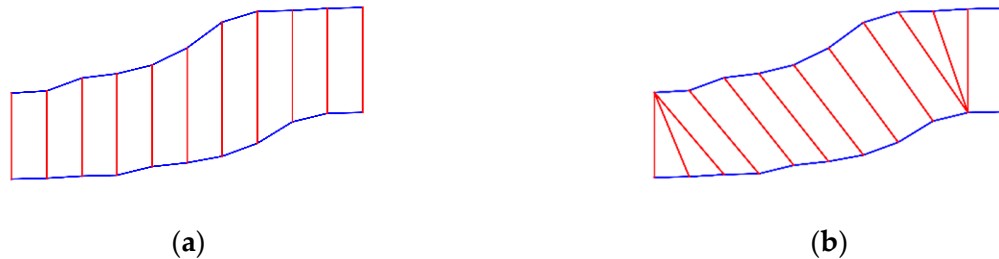

(**a**)                                    (**b**)

**Figure 5.** Matching patterns of Euclidean distance and dynamic time warping (DTW). (**a**) Matching based on trace points by Euclidean distance; (**b**) matching based on trace points by DTW.

The computing method of DTW distance is as follows: for time series $X = [x_1, x_2, \ldots, x_i, \ldots, x_m]$ and $Y = [y_1, y_2, \ldots, y_j, \ldots, y_n]$, in which $m$ and $n$ can be unequal, DTW seeks a warping path $W$ to represent the matching relationship between $X$ and $Y$. $W = [w_1, w_2, \ldots, w_k, \ldots, w_K]$, in which $\max(n, m) \leq K \leq n + m - 1$. The $k$th element of $W$ is marked as $w_k = (i, j)$, which represents the correspondence between the $i$th element of $X$ and the $j$th element of $Y$. There are three constraint conditions for the selection of the warping path: (1) the warping path starts with the first element of the matrix and ends with its diagonal element, which means $w_1 = (1, 1)$ and $w_K = (m, n)$; (2) each step of the warping path is continuous, which means that for $w_k = (a, b)$ and $w_{k-1} = (a', b')$, they should satisfy $a - a' \leq 1$ and $b - b' \leq 1$; (3) the warping path is monotonous on the time dimension, which means that for $w_k = (a, b)$ and $w_{k-1} = (a', b')$, they should satisfy $a - a' \geq 0$ and $b - b' \geq 0$.

There will be various paths that satisfy the constraint conditions, and the path with the lowest warping cost is the target one, which is:

$$D_{DTW}(X, Y) = \min\left\{ \sqrt{\sum_{k=1}^{K} d(w_k)} \right\} \tag{2}$$

where $d(w_k)$ is the distance between the two elements represented by $w_k$.

According to the theory of dynamic planning, if element $(i, j)$ is located on the best path, the subpaths from $(1, 1)$ to $(i, j)$ are local optimal paths. Therefore, the best path can be obtained by recursively searching the local optimal paths between $(1, 1)$ and $(m, n)$, which can be realized by the following steps: first, build a matrix of $m \times n$ order with element $(i, j)$ being the distance between point $x_i$ and $y_j$, which can be calculated by $d(x_i, y_j) = (x_i - y_j)^2$; second, calculate the cumulative distance of element $(i, j)$ by equation:

$$\gamma(i, j) = d(x_i, y_j) + \min\{\gamma(i-1, j-1), \gamma(i-1, j), \gamma(i, j-1)\} \tag{3}$$

The cumulative distance matrix can be computed by recursive calculation with an initial condition of $\gamma(1, 1) = d(x_1, y_1)$. The DTW distance of time series $X$ and $Y$ is $\sqrt{\gamma(m, n)}$. The best path can be acquired by reverse searching from point $(m, n)$, as shown in Figure 6.

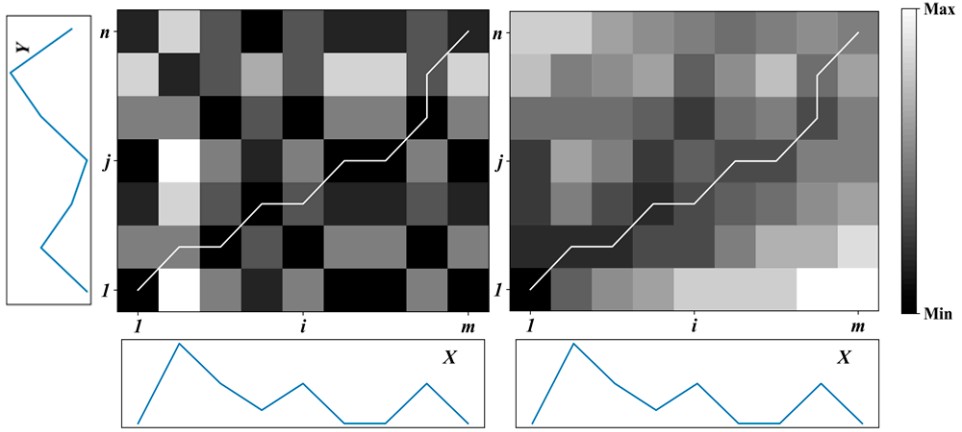

(**a**) Matrix of element distance　　(**b**) Matrix of cumulative distance

**Figure 6.** Calculation matrix and warping path of DTW.

Another key factor of time series clustering is the logic to generate clusters, which can be summarized as partitioning clustering such as k-means [43], hierarchical clustering such as BIRCH [44] and BHC [45], model-based clustering such as HMMs [46] and ARMA-EM [47] and density-based clustering such as DBSCAN [48]. Among these algorithms, hierarchical clustering has no requirement for the initial determination of the number of clusters, which is an outstanding advantage in rainfall pattern analysis because it is hard to define the number of rainfall temporal patterns before the analysis.

A simple hierarchical clustering method named AGNES (agglomerative nesting) was used in this study. The algorithm steps are as follows:

(1) Each element in the time series set is taken as an initial cluster. For a set with $m$ elements $D = \{x_1, x_2, \ldots, x_m\}$, the initial cluster $C = \{C_1, C_2, \ldots, C_m\}$, in which $C_j = \{x_j\}$;

(2) Calculate the distance matrix $M$, whose element $M(i, j) = d(C_i, C_j)$ and $M(i, j) = M(j, i)$, $d(C_i, C_j)$ is the distance between cluster $C_i$ and $C_j$;

(3)   Find the closest two clusters $C_{i*}$ and $C_{j*}$ and merge them: $C_{i*} = C_{i*} \cup C_{j*}$. Renumber the clusters and delete the *j*th row and *j*th column of matrix *M*;

(4)   Repeat the previous step until all clusters merge into one cluster, and then a clustering tree is obtained.

A potential limitation of this algorithm is the high computational complexity when dealing with large sample sets. The time complexity of the DTW algorithm is O($m \times n$), where *m* and *n* are the length of the time series, and the time complexity of the first matrix *M* is O($N \times N$), where *N* is the number of elements in the time series set, which means the time cost has an exponential growth in the calculation of the first matrix. A considerable number of approaches have been proposed to improve the computational efficiency of DTW, such as reducing the dimension of time series [49–53], setting boundary constraints and narrowing the scope of path searching [42,54], and setting the lower bounding function to reduce the computing times of DTW [55–58]. All these methods mentioned above make local improvement in the algorithm and have limited effects on efficiency improvement. Since the calculation of each element in the first matrix is independent, GPU parallel computing was used in this study to accelerate the calculation of the first distance matrix. Each matrix element was assigned with a GPU thread for the calculation of DTW distance; therefore, the matrix computation changed from loop computing to parallel computing, which can significantly improve the computing efficiency when dealing with large matrices. Threads per block and blocks per grid should be defined before calling the GPU. The number of threads within a thread block is limited by the GPU device in use, and the number of thread blocks within a thread grid is determined by the number of rainfall samples.

### 2.2.3. Extraction of Representative Temporal Patterns

The representative temporal patterns of local rainstorms were extracted based on the clustering tree obtained above. The root node of the clustering tree is treated as the first layer, and thus there are *n* nodes in the *n*th layer. Each node of the clustering tree represents a cluster, which means a group of similar rainfall processes. The objective of pattern extraction is to separate different rainfall processes as much as possible, which means more divided clusters. There should be sufficient samples in each cluster to ensure that the extracted events are representative enough. Therefore, the number of representative temporal patterns should be determined according to two factors: the total number of clusters and the number of events in each cluster. The event with the smallest sum of distances from other events in one cluster is treated as the clustering centre, which is the representative temporal pattern extracted from this cluster.

## 3. Results and Discussion

### 3.1. Representative Temporal Patterns

Clustering analysis and the extraction of representative temporal patterns were conducted based on rainstorm events belonging to each duration section. The clusters and temporal patterns corresponding to the duration section [3, 6) are shown in Figure 7. The number of clusters was chosen as five so that the events in each cluster are relatively evenly distributed (see Table 2), which ensures the diversity and representativeness of the temporal patterns. There are remarkable characteristics of each cluster as well as differences between distinct temporal patterns: (a) temporal pattern I represents a smooth rainfall process whose peak value is not significant and the main process lasts for the whole rainfall duration; the distribution of intensity in the process is similar to normal distribution; (b) temporal pattern II represents a rainfall process that has little rainfall before the peak, which means the peak usually appears abruptly, and the main process lasts for more than one hour but does not cover the whole duration; (c) the characteristics of temporal pattern III are that the peak value is the main contribution of rainfall depth, and there are small amounts of rainfall before the peak value and little rainfall after the peak appears; (d) similar to temporal pattern III, the total rainfall depth of temporal pattern IV also concentrates in one hour, but the difference is that there is little rainfall before the peak value and

small amounts of rainfall after the peak appears; (e) the total rainfall depth of temporal pattern V is also concentrated in one hour and has little rainfall in other time periods; and (f) temporal patterns whose total rainfall depths concentrate in one time interval (temporal pattern III, IV and V) account for approximately 54.8% of the total events.

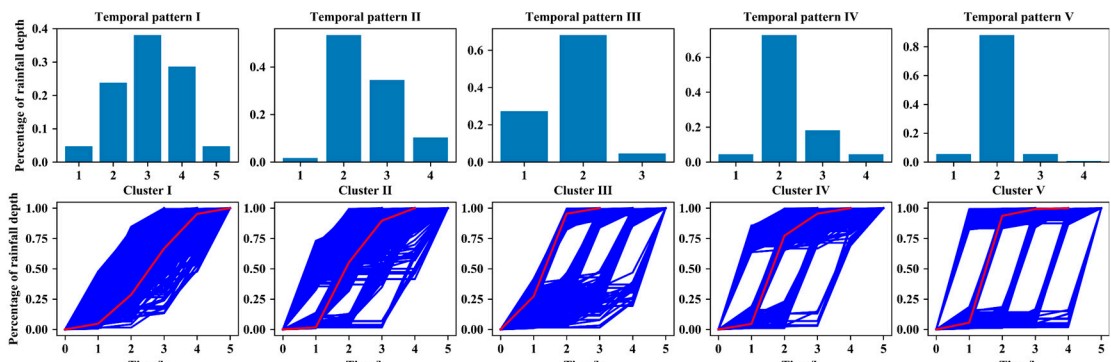

**Figure 7.** Temporal patterns and clusters extracted from section [3, 6).

**Table 2.** Cluster characteristics of each duration section*.

| Duration Section | Clusters | No. of Events (Percentage) | Intensity /mm | Peak Value /mm | Rainfall Depth /mm | Ratio of Peak Value to Rainfall Depth |
|---|---|---|---|---|---|---|
| [3, 6) | Cluster I | 432 (17.9%) | 3.8 | 7 | 15.5 | 41.70% |
| | Cluster II | 658 (27.2%) | 4.8 | 10 | 18 | 54.50% |
| | Cluster III | 334 (13.8%) | 5.2 | 13.1 | 20 | 65.80% |
| | Cluster IV | 454 (18.8%) | 4.8 | 13.5 | 18 | 73.90% |
| | Cluster V | 537 (22.2%) | 5.1 | 16 | 18 | 88.70% |
| [6, 12) | Cluster I | 1805 (44.7%) | 2.3 | 6.5 | 19 | 34.8% |
| | Cluster II | 1023 (25.3%) | 2.4 | 9 | 20.4 | 44.8% |
| | Cluster III | 342 (8.5%) | 2.4 | 13 | 19.8 | 67.8% |
| | Cluster IV | 554 (13.7%) | 2.6 | 12 | 20 | 63.3% |
| | Cluster V | 312 (7.7%) | 2.6 | 16.5 | 19 | 85.7% |
| [12, 24) | Cluster I | 2775 (64.8%) | 1.5 | 6 | 24.5 | 23.80% |
| | Cluster II | 854 (19.9%) | 1.7 | 11 | 26.8 | 38.70% |
| | Cluster III | 150 (3.5%) | 1.8 | 14.5 | 25.8 | 56.10% |
| | Cluster IV | 438 (10.2%) | 1.7 | 10.5 | 25.5 | 42.10% |
| | Cluster V | 64 (1.5%) | 1.8 | 19.8 | 24.8 | 78.60% |
| [24, 48) | Cluster I | 1828 (84.4%) | 1.3 | 7 | 43 | 16.50% |
| | Cluster II | 208 (9.6%) | 1.6 | 15 | 46.3 | 30.80% |
| | Cluster III | 74 (3.4%) | 1.2 | 12 | 37.8 | 33.60% |
| | Cluster IV | 39 (1.8%) | 1.2 | 17.5 | 36 | 45.60% |
| | Cluster V | 16 (0.7%) | 1.5 | 18.8 | 42.4 | 55.70% |
| [48, 96) | Cluster I | 320 (84%) | 1.5 | 9.2 | 92.2 | 10.3% |
| | Cluster II | 36 (9.4%) | 1.5 | 12.8 | 75 | 17.2% |
| | Cluster III | 25 (6.6%) | 1.8 | 19 | 105 | 20.9% |

\* All statistical values are median values.

Five temporal patterns similar to those corresponding to duration section [3, 6) were extracted based on events in section [6, 12), which include a smooth process without notably high values; a process lasting for more than one hour with the peak appearing abruptly; a process in which the rainfall depth concentrates in one hour with small amounts of rainfall before the peak value; a process in which the rainfall depth concentrates in one hour with small amounts of rainfall after the peak value; and a process in which the rainfall depth concentrates in one hour with little rainfall in other time periods, as shown in Figure 8. The proportion of the five patterns changes with a significant increase in temporal pattern I and a decrease in temporal patterns III, IV and V (see Table 2), which means that the events in which the total rainfall depths concentrate in a single time interval become less frequent and signifies the processes of rainstorms becoming more complicated with increasing duration. In addition, the processes of events in Cluster I become more diverse and deviate from normal distribution (see Figure 8).

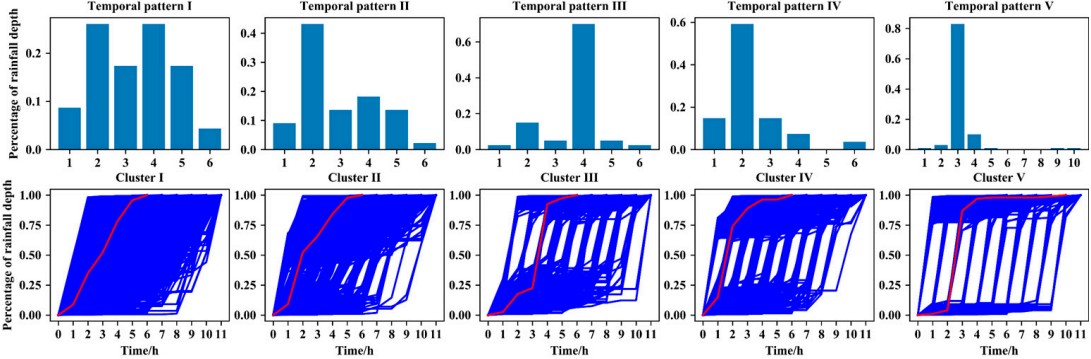

**Figure 8.** Temporal patterns and Clusters extracted from section [6, 12).

For duration sections [12, 24) and [24, 48), the clusters and extracted patterns still have similar features to those of the previous sections (see Figures 9 and 10). However, there are also some variations: (a) the number of events belonging to Cluster I increases significantly and the proportion of Cluster I achieves 84.4% for section [24, 48); (b) the processes of events in Cluster I become more complex and diversified, and the representative pattern presents a relatively uniform process for section [12, 24) and a bimodal process for section [24, 48); (c) for both sections, the representative patterns of Cluster II exhibit a process quite similar to the Chicago hyetograph proposed by Keifer and Chu [27]; and d) even though the events with total rainfall depths concentrated in one time interval decrease significantly, there are still a certain proportion of rainfall processes belonging to temporal patterns III, IV and V (see Table 2).

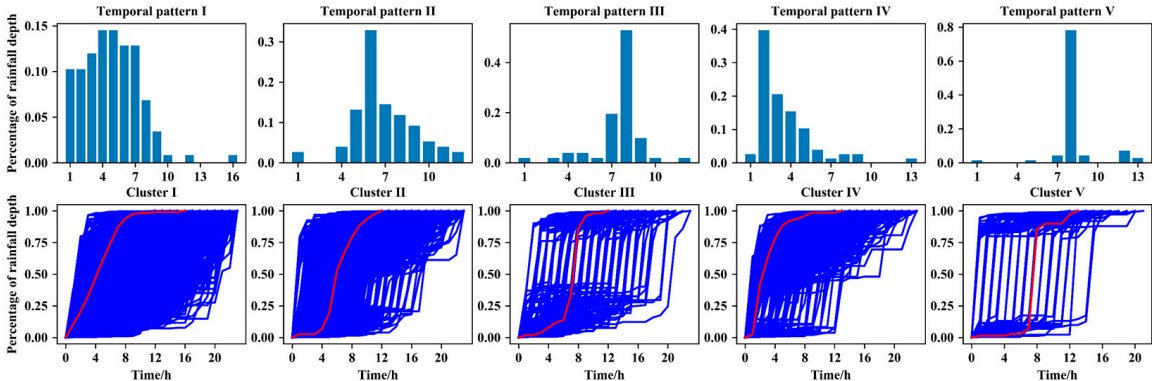

**Figure 9.** Temporal patterns and clusters extracted from section [12, 24).

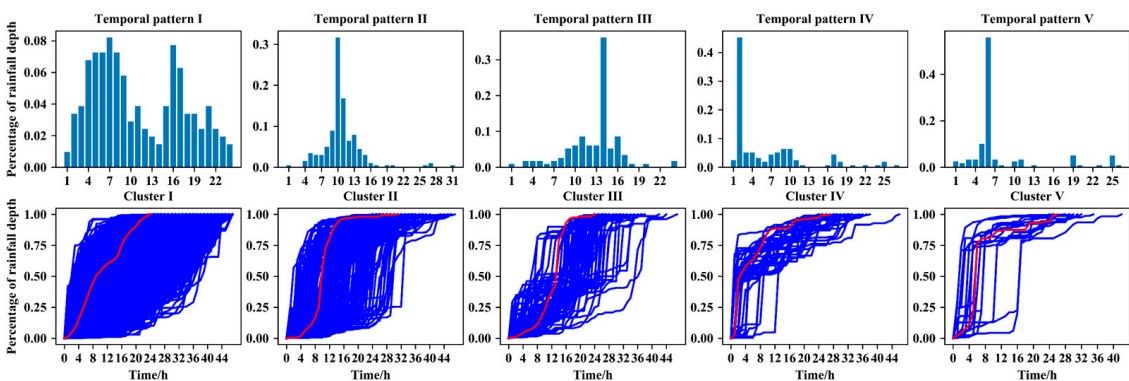

**Figure 10.** Temporal patterns and clusters extracted from section [24, 48).

As the number of events in section [48, 96) is limited, the number of clusters is chosen as three to ensure the representativeness of the extracted temporal patterns. As shown in Figure 11, the three temporal patterns are relatively complex, and their characteristics can be roughly described as follows: a complex process with multiple and unobvious peaks; a process that has small amounts of rainfall before and after the peak value; and a process that has small amounts of rainfall before the peak value and small amounts of rainfall after the peak value. For the duration section [96, 192), there are only 21 events, which are not enough to support the cluster analysis. Therefore, we will not conduct further analysis for this duration section. We can see that the number of samples and the complexity of rainfall processes can influence the effect of clustering analysis. For sample events with short durations and large sample numbers for analysis, the extracted temporal patterns are relatively representative, but for sample events with long durations and small sample numbers, it is hard to extract representative patterns.

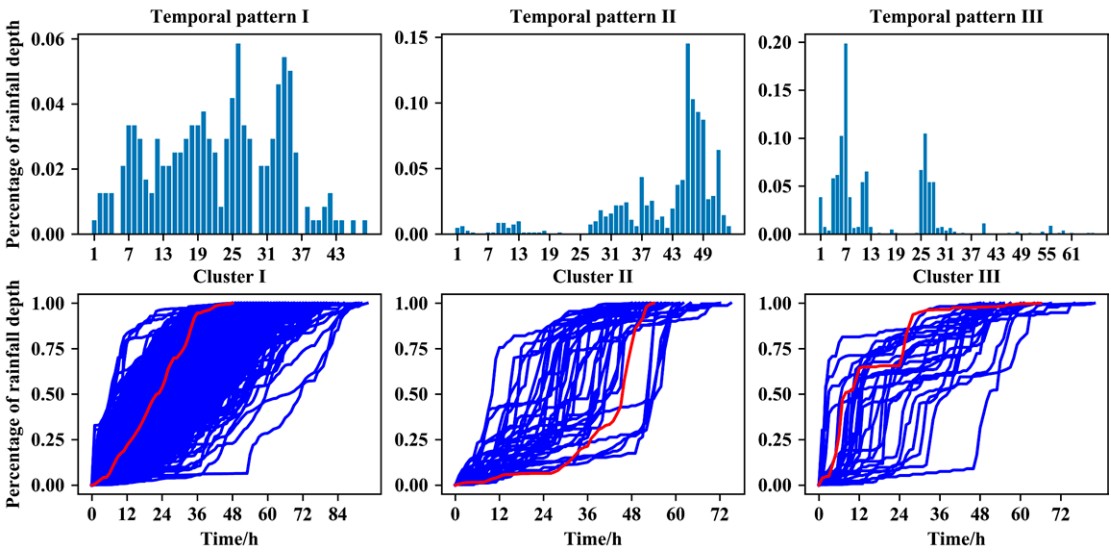

**Figure 11.** Temporal patterns and clusters extracted from section [48, 96).

### 3.2. Statistical Characteristics of the Clusters and Temporal Patterns

We can find more features of rainstorms by statistical analysis and the comparisons of different clusters and temporal patterns within and between duration sections.

The distribution of the peak value location (Figure 12) shows that for Cluster III, the number of events whose peak value is located in the first time interval is zero, which means there are always small amounts of rainfall before the peak appears and verifies the previous description of temporal pattern III. It also shows that for temporal patterns I, II, IV and V, the peaks are more likely to occur in the early

part of the events than in the late part of the events. This feature is especially significant for temporal pattern IV. However, for temporal pattern III, the peaks are more inclined to occur in the middle part of the events. The distribution of rainfall durations (Figure 13) shows that there is a slight reduction in the number of events with the increasing of rainfall duration for temporal patterns II, III and IV, and it is basically evenly distributed. Significant differences should be noted in temporal patterns I and V. For temporal pattern I, the duration continues to increase in section [3, 12) and decreases in section [12, 48), which implies that for this type of rainstorm, a duration of 12 h is the most common situation. For temporal pattern V, the number of events decreases significantly with increasing event duration, which implies that this type of rainstorm tends to be an event with a short duration.

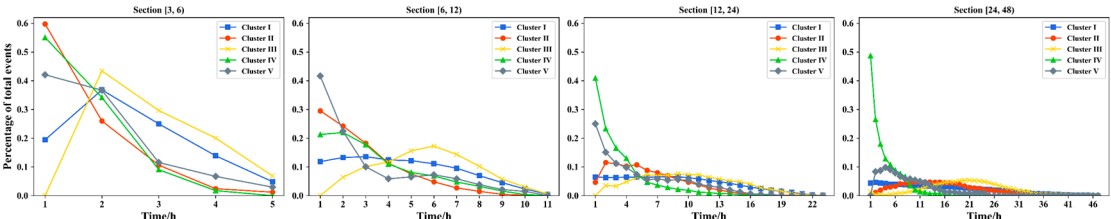

**Figure 12.** Distribution of peak value location.

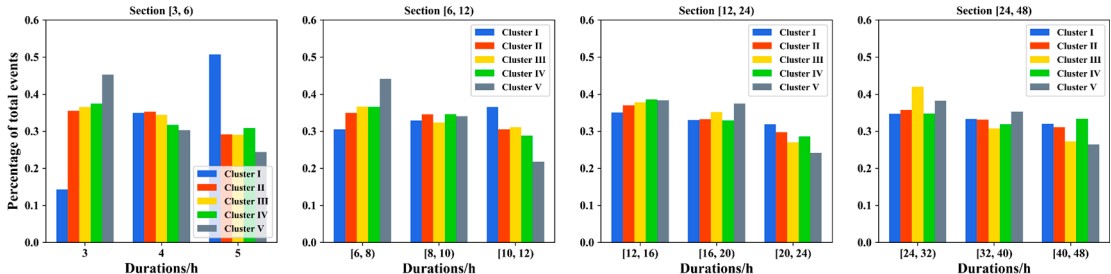

**Figure 13.** Distribution of rainfall durations.

From the statistical characteristics of clusters and duration sections (see Table 2 and Figure 14), we can see that: (a) for each duration section, there are no significant differences in the statistical characteristics of rainfall intensity and rainfall depth between clusters; however, for peak value, the difference is obvious, with Cluster I having the smallest peak value and Cluster V having the largest one, which is basically more than twice that of Cluster I; this statistical regularity is in accordance with the features of the temporal patterns described in the previous subsection, which states that the rainfall depths of temporal pattern V basically concentrate in one time-interval; (b) the same feature is also shown by the statistics of the ratio of peak value to rainfall depth, which shows that Clusters I and V always have the smallest and largest one respectively; (c) the similarity of rainfall depths between clusters suggests that temporal patterns III, IV and V are more dangerous for flood control because, even though the main processes of these temporal patterns are shorter than those of temporal patterns I and II, there is basically no reduction in rainfall depths, and it is a consensus that more concentrated rainfall intensity usually brings larger runoff amounts; (d) even though the peak values of each cluster within one duration section have differences, the peak values of each cluster between different duration sections are basically the same, which means the extension of durations does not bring larger peak values; (e) the rainfall depths are increasing with the extension of durations, but for duration sections [3, 6) and [6, 12), the growth is not obvious, which implies that the total rainfall depths of rainstorm events with durations within 12 h mainly concentrate in 3 h, or even 1 h; (f) the same situation exists for duration sections [6, 12) and [12, 24), which shows that the rainfall depths only increase by approximately 25% for another 12 h of duration, and the main processes are mainly within 6 h; (g) furthermore, the variation patterns of rainfall depths and intensities change when the rainfall durations are beyond 24 h, which shows that the rainfall depths and intensities of time periods other

than the peak noticeably increase; (h) events lasting longer than 48 h are more similar to combinations of several short-duration rainfall processes because the intensities are close to those of section [12, 24].

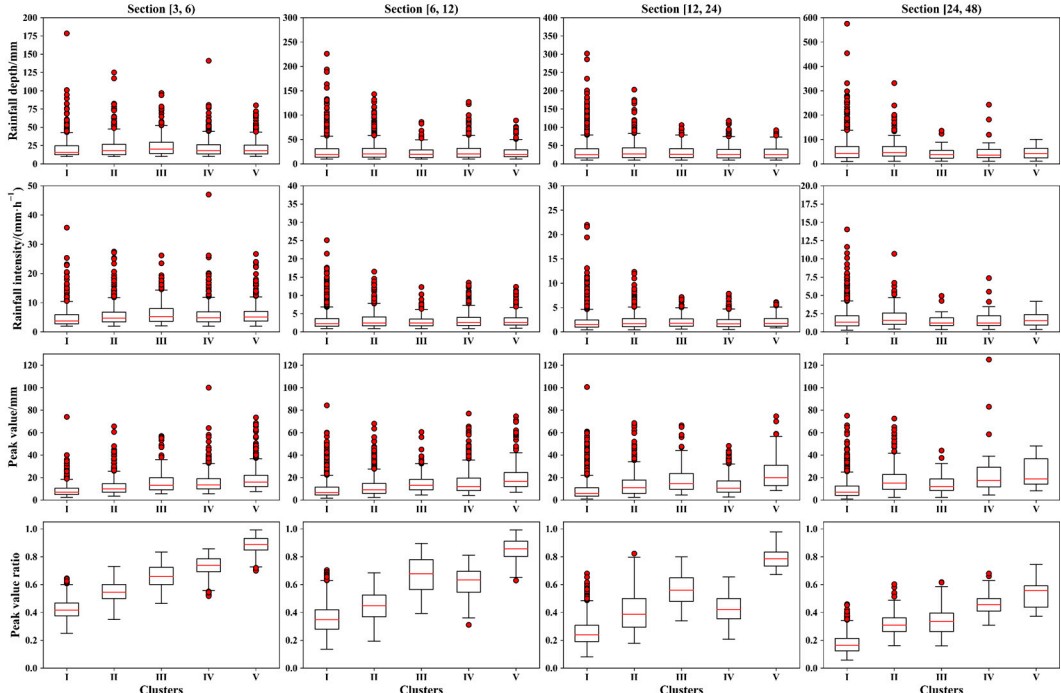

**Figure 14.** Distribution of rainfall depth, intensity, peak value and peak value ratio.

We can see that there exist significant differences between the statistical characteristics of each cluster and the representative temporal pattern, which corroborates their features and validates their representativeness.

### 3.3. Regional Analysis of Rainstorm Characteristics

Due to the significant climate diversities in different regions in China, further analysis has been performed to find out the differences and the similarities of clusters and temporal patterns between geographical zones. However, from the pie chart (see Figure 15) we can find that there seems to be no significant difference in temporal patterns between regions. For each duration section, the proportions of clusters in different geographical zones are almost the same, and the maximum difference in the proportion of a single rainfall temporal pattern between regions does not exceed 9%. This shows that the divided clusters and the extracted rainfall temporal patterns have no obvious regional characteristics, and it implies that the macroclimate does not play a decisive role in the temporal patterns of local rainstorms. However, it is valuable to conduct further studies on the influences of local climate and topography on rainfall temporal patterns with the support of abundant data.

Furthermore, the distributions of the rainfall characteristic values of each geographical zone were calculated in accordance with the duration sections. As shown in Figure 16, the quantiles of rainfall depths, intensities and peak values in South China (S), Central China (C) and East China (E) are higher than the overall average values, and the quantiles of those values in Northeast China (NW), North China (N), Northwest China (NW) and Southwest China (SW) are lower than the average values. This indicates that the areas greatly affected by the southeast monsoon have stronger rainstorm events, which is consistent with the regular pattern that the rainfall decreases from southeast to northwest in China.

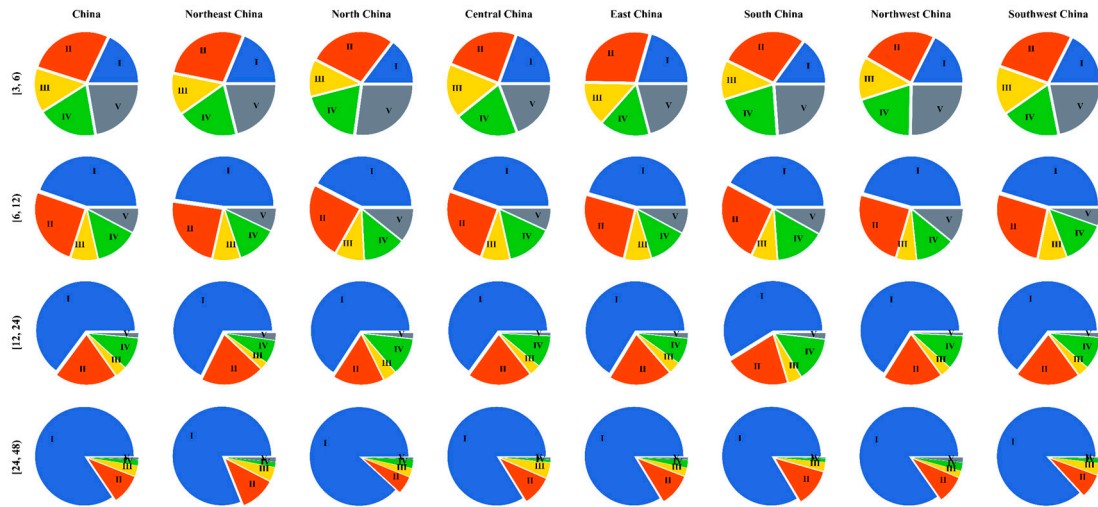

**Figure 15.** Proportion of clusters in different time sections and geographical zones.

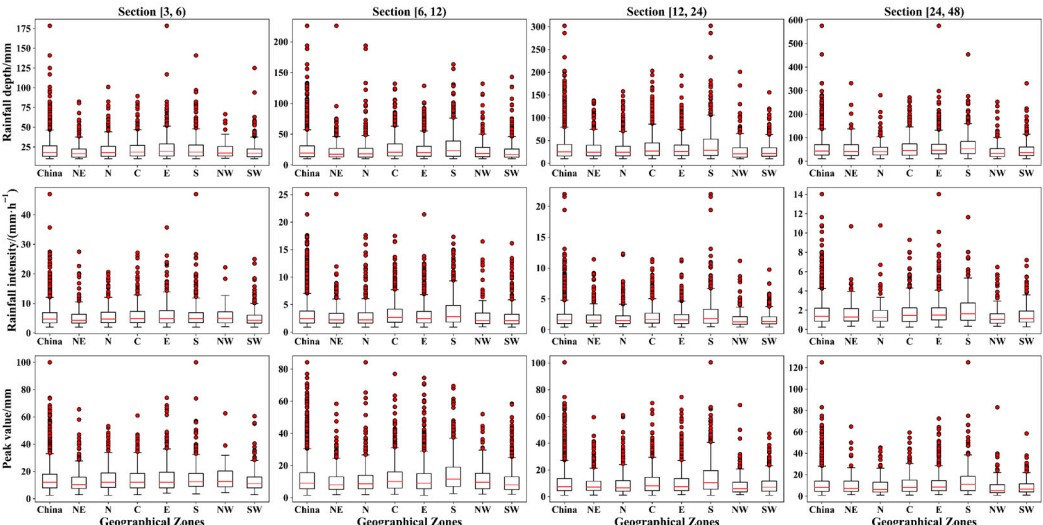

**Figure 16.** Distribution of rainfall characteristic values in different geographical zones.

## 4. Conclusions

A total of 13,299 rainstorm events from 99 hydrological and precipitation stations were analyzed using time series clustering to explore the temporal patterns of local rainstorm events during the flood season in China.

The total rainstorm events were divided into different groups based on their durations. Then, the events in the same group were clustered according to the similarity criteria, and the representative temporal patterns were extracted from each cluster. The results show that for rainstorm events with durations of less than 48 h, there are five representative temporal patterns in each duration section, which are characterized by: pattern I, a smooth process without notably high values; pattern II, a process lasting longer than one hour and the peak appearing abruptly; pattern III, a process in which the rainfall depth concentrates in one hour with small amounts of rainfall before the peak value; pattern IV, a process in which the rainfall depth concentrates in one hour with small amounts of rainfall after the peak value; and pattern V, a process in which the rainfall depth concentrates in one hour with small rainfall amounts in other time periods. These short-duration temporal patterns can be treated as the five most representative ones of local rainstorm events in China during the flood season.

In addition, the statistics and comparisons of events within and between clusters show that: (a) compared with other temporal patterns, the peaks of temporal pattern III appear relatively posteriorly, which implies that it is more likely to cause flooding because the soil moisture is higher when the peaks appear; (b) the rainfall depths of each cluster have no obvious differences, which indicates that temporal patterns III, IV and V are more dangerous for flood control because they have more concentrated rainfall within the processes; and (c) for rainstorm events whose durations are less than 12 h, their rainfall mostly concentrates in 3 h, and for rainstorm events whose durations are within 24 h, the rainfall mostly concentrates in 6 h; this indicates that even though the total rainfall durations are long, the disastrous rainfall processes usually occur in quite a short time, which places the main pressures on local flood control and needs extra attention. Compared with using average intensities of 12 or 24 h as design standards, this proposes higher requirements for the planning and design of flood control and drainage projects.

The study shows that the time series clustering method using DTW as similarity measurement criteria is effective in rainfall temporal pattern analysis. However, for rainstorm events with durations longer than 48 h, the complex processes and limited samples will influence the effect of clustering. It can be improved by increasing the time step and enlarging the sample sizes. However, this shows its shortcomings; that is, the demand for data sizes is relatively high.

Through the regional analysis of rainstorm characteristics, it can be seen that the rainfall depth, intensity and peak value are regional characteristics. This means that they are affected by the macroclimate; in this case, it is the southeast monsoon during the flood season. However, the temporal patterns do not show obvious differences between geographical zones, which indicates that it is not strongly related to the macroclimate but more likely to be affected by the local climate and topography, which needs further studies with smaller scale data and more detailed contrastive analysis.

**Author Contributions:** Conceptualization, F.W.; methodology, F.W.; validation, F.W.; writing—original draft preparation, F.W.; writing—review and editing, F.W. All authors have read and agreed to the published version of the manuscript.

**Funding:** This research was funded by the National Key R&D Program of China (2017YFC1502602) and the IWHR Research & Development Support Program (JZ0145B772017, JZ0145B022019).

**Conflicts of Interest:** The authors declare no conflict of interest.

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
