# Peer review of "Temporal Pattern Analysis of Local Rainstorm Events in China During the Flood Season Based on Time Series Clustering"

_water, doi:10.3390/w12030725_

Round 1

Reviewer 1 Report

This is interesting paper. It can be accepted once the below minor comments are addressed:

The literature review needs to be extended. Two relevant papers have not been cited: (a) Rahman, A., Islam, M., Rahman, K., Khan, S. and Shrestha, S. (2006). Investigation of Design Rainfall Temporal Patterns in the Gold Coast Region Queensland. Australian Journal of Water Resources, 10, 1, 49-61. (b) Rahman, A., Weinmann, P. E., Hoang, T.M.T, Laurenson, E. M. (2002) Monte Carlo Simulation of flood frequency curves from rainfall. Journal of Hydrology, 256 (3-4), 196-210. Discuss the stochastic nature of temporal patterns as mentioned in the above papers. Please state the quality of the rainfall data used in your study. Please state the practical applicability of the temporal patterns you have generated. Compare your results with similar studies (already published in journals).

Author Response

Thank you for your generous comments. I've studied the two relevant papers you recommended and cited them because they are inspiring works. It provides an ingenious approach to combine temporal patterns into design flood estimation. The rainfall data used in my study is from the ministry of water resources, which is reliable. The generated temporal patterns have not been used in pratical cases yet, but I hope this method can be adopted in more cases such as determining flash flood thresholds.

Reviewer 2 Report

In the introduction it would be worth mentioning the areas (cities) in China which are particularly vulnerable to occurrence of flash flood and urban flood inundation and their frequency. Line 139: The duration of selected sections was used in other publications? Does the rainfall data used in the study come from precipitation stations located in the mountains? If so then it should be mentioned in the text. In this case, Author of  the article should also refer to temporal distributions of rainfall during rainstorm events at mountains and lowlands, whether they are similar or not. Figures 7-11 It is recommended  to use equal values of extremes on the axis y (percentage of rainfall depth) for comparative purposes - at least for data with a similar range. I am aware that this is a problem is related to significant differences of extremes. The same recommendations for rainfall depth and rainfall intensity (Figure 14) and Figure 4. Very often sentences start with a word “meanwhile”. I would suggest replacing the word with a different one or slightly change the sentence structure. Line 288 Change the word “decease” to “decrease”. Line 398 Change the word “china” to “China

Author Response

Thank you for your generous comments. The duration sections used in this study were determined by the durations of total events I have extracted from the original rainfall data. I divided the whole duration period into several sections to make the distribution of events uniform. I did not find any publication using this division method because most of them used fixed duration or dimensionless curves. The landforms of the gauges have not been considered in this study because I have not collected detailed information of these gauges. But I think your advises are valuable, as I have mentioned in the article, the temporal patterns may be impacted by the local landform and climate. I think it is valuable to conduct comparision study between temporal patterns of mountain regions and lowlands with more detailed data. I have edited the language once again and sorry for the spelling mistakes.